# Comparative analysis of patient-reported outcomes in joint arthroplasty surgeries

Ville Äärimaa[1,2☯], Karita Kohtala[ID][2☯]*, Keijo Mäkelä[1,2‡], Mikko Karvonen[1,2‡], Anssi Arimaa[1,2‡], Anssi Ryösä[1,2‡], Joel Kostensalo[ID][3], Fanny Kaivonen[1], Inari Laaksonen[1,2☯]

1 Department of Orthopedics and Traumatology, Turku University Hospital, Turku, Finland, 2 University of Turku, Turku, Finland, 3 Natural Resources Institute Finland, Joensuu, Finland

☯ These authors contributed equally to this work.
‡ KM, MK, AA and AR also contributed equally to this work.
* kshkoh@utu.fi

## Abstract

### Background

This study aims to report and analyze disease-specific patient-reported outcome measure (PROM) effect size (ES) variations, in patients undergoing major arthroplasty surgery.

### Material and methods

All institution-based data of primary knee, hip, or shoulder arthroplasty patients at Turku University hospital (Finland) between January 2020 – December 2022 were collected, and treatment outcome assessed as a PROM difference between baseline and one-year follow-up. PROM ES were calculated for each patient and patient group separately, and patients with ES >0.5, were considered responders. Factors contributing to patient outcome and differences between patient groups were investigated using linear models and non-parametric methods.

### Results

2580 patients were operated (complete follow-up data on 1828 patients). 1110 (61%) of the patients were female, and mean age was 69 years (SD 10). The mean ES across all patient groups was 2.64 (SD 1.29) and the biggest ES was observed in shoulder patients and the smallest in knee patients. Smaller ES was statistically significantly associated with higher preoperative PROM, higher ASA class, and old age. The percentage of responders was highest for shoulder patients (97.7%), followed by hip patients (96.8%), and lowest for knee patients (92.5%).

### Conclusion

The observed ES for joint arthroplasty surgeries is high. However, there are significant disparities among patients with primary knee, hip, and shoulder joint arthroplasty surgery. These variations are mainly due to differences in preoperative PROM score and may be attributed to differences in patient selection. We recommend that prior to shared decision-

**Citation:** Äärimaa V, Kohtala K, Mäkelä K, Karvonen M, Arimaa A, Ryösä A, et al. (2024) Comparative analysis of patient-reported outcomes in joint arthroplasty surgeries. PLoS ONE 19(12): e0314818. https://doi.org/10.1371/journal.pone.0314818

**Data Availability Statement:** Sharing the raw data in its entirety is not possible due to compliance with GDPR regulations and need to protect sensitive personal information. Key summary

statistics and aggregated data can be shared upon reasonable request. Interested researchers can contact the Auria Clinical Informatics, tutkimuksentietopalvelut@varha.fi, 20521 Turku, Finland, for further details.

**Funding:** This research was funded by government research funding. The funder did not have any role in the design or production of the study or the manuscript.

**Competing interests:** Ville Äärimaa has received grants from the Academy of Finland, the Social Insurance Institution of Finland, and the Turku University hospital. Ville Äärimaa has received payment for expert testimony given to National Patient Injury Board. This does not alter our adherence to PLOS ONE policies on sharing data and materials. Inari Laaksonen have received grants from the State Research Funding of Southwestern Finland. Turku University Hospital has supported Ville Äärimaa, and Inari Laaksonen to attend meetings and/or travel.

making, preoperative scores are thoroughly reviewed with the patient, along with other patient specific factors that may influence the end result of the treatment.

## Introduction

Over 500 million people worldwide suffer from osteoarthritis with varying and progressing symptoms of pain and locomotive impairment [1]. With the ageing population, rising demand for treatment, and the prevalence of obesity, there is a noticeable increase in the number of knee, hip, and shoulder joint arthroplasty surgeries performed [2].

Patient-reported outcome measures (PROMs) have emerged as important tools for evaluating patient health, monitoring changes, and assessing treatment outcomes, particularly in degenerative musculoskeletal diseases [3]. A clinically relevant change in a validated disease-specific PROM offers the most precise and responsive indication of a treatment effect detectable by patients themselves [4]. Despite the increasing number of studies reporting PROM outcomes in specific arthroplasty procedures, there remains limited understanding of how these outcomes compare across different anatomical regions of arthroplasty surgery. Moreover, the comparative patient-reported treatment effects and factors affecting it are obscure. Ultimately, it is in the interest of every patient and physician to comprehend the expected treatment outcome before shared decision making on elective operative treatment [3, 5].

In this study, our goal was to examine and report the outcomes and effect sizes (ES) of disease-specific PROMs in primary knee, hip, and shoulder arthroplasty surgery for osteoarthritis within a comprehensive institutional PROM registry. Our specific objective was to examine differences, variation, and reasons for variation in PROM ES in these patient groups.

## Materials and methods

An institution-based anatomically divided systematic PROM registry for all patients undergoing joint arthroplasty surgery was established in 2019 at Turku University Hospital (Finland) with a catchment area of ca. 500,000 people. The registry was comprised of anatomic subregistries with disease-specific PROM outcome instruments as presented in Table 1. The scores were gathered preoperatively at baseline and at one year postoperatively using an electronic application (Omavointi, BCB Medical, Turku, Finland). For this study, all PROM data were scaled to range from 0 to 100 and harmonized in a way that a positive change in the score represented an improvement in the perceived quality of life. Patient history data were gathered regarding age, sex, American Society of Anesthesiology (ASA) score, and body mass index (BMI). Surgery related data on the main diagnosis code (according to the International Classification of Diseases (ICD) 10 classification), and the main operation (according to the Classification of Surgical Procedures Version 1.14 by the Nordic Medico-Statistical Committee (NOMESCO)) were also gathered (see S1 Table). The data were accessed 2.4.2024.

Patients with end stage osteoarthritis and primary joint arthroplasty surgery for knee, hip, or shoulder were included using operative coding and International Classification of Diseases

**Table 1. Patient groups and disease-specific patient-reported outcome measures (PROM).**

| Patient group | PROM | previously reported MCID (minimal clinically important difference) | reference |
|---|---|---|---|
| Shoulder arthroplasty | Western Ontario Osteoarthritis of Shoulder (WOOS) | 12.3 (12.3%) | [6] |
| Hip arthroplasty | Oxford Hip Score (OHS) | 5.2 (10.8%) | [7] |
| Knee arthroplasty | Oxford Knee Score (OKS) | 4.7–10 (9.8–20.8%) | [8] |

(ICD-10) codes. All patient data between January 1$^{st}$, 2020 and December 31$^{st}$, 2022 were collected, and the treatment outcome assessed as a difference between baseline and one-year follow-up PROM values. To compare results between groups we adopted a unified measure of outcome which we refer hereafter as ES. This measure can be understood as a standardized outcome measure. The effect size corresponding to observation $i$ is given here by

$$ES_i = \frac{PROM_{12\ months,i} - PROM_{preop,i}}{\text{SD}(PROM_{preop})}$$

where $PROM_{12\ months,i}$ is the PROM response 12 months after the operation for the patient $i$, $PROM_{preop,i}$ is the preoperative PROM-score of the patient, and $\text{SD}(PROM_{preop})$ is the standard deviation of all the PROM-scores in the registry. The patients with $ES_i > 0.5$ are referred to as *responders*, i.e., individuals for whom the surgical treatment has led to an improvement over the distribution-based estimate of minimally clinically important difference (MCID) [9]. Statistical differences in the fractions of positive responders between anatomical groups were tested using Fisher's exact test. Statistical significance of the differences in ES and unnormalized change in PROM between anatomical groups were tested using Kruskal's test.

The differences between groups were analyzed using a linear model, in which ASA class, anatomical group, age, BMI, and preoperative PROM were covariates. Based on graphical checks, the effect of age was modeled as constant up to 65 and linearly for older patients. Six BMI outliers suspected of being erroneous (BMI = 53–2700, with the distribution being continuous up to 47) were removed from the linear model fit. An alternative model with sex included as covariate was explored, but the association of sex with effect size was not statistically significant and based on the Akaike Information Criterion (AIC) a model without it was preferred. As a sensitivity analysis we also fitted an identical model with the unnormalized change in PROM as the response variable. Factors contributing to a patient being a responder (ES>0.5) were tested using an analogous generalized linear model with a binary response variable with a logit link function. The regression coefficients for the latter two models are given in the Supplementary material (S2 and S3 Tables).

All statistical analyses were carried out using the statistical software R [10].

Permission for this study was obtained from the Internal Review Board of Turku University Hospital (TurkuCRC). The study has been performed in accordance with the ethical standards of 1964 Declaration of Helsinki. Our research permit number is T283/2020 and institutional approval was granted 18.2.2021. Ethical approval was not required in our institute as this study is registry based retrospective analysis and the study had no effect on patients' treatment. All patient data was deidentified during analyses.

## Results

The demographic variables are presented in Table 2. In total, 2580 patients were operated (complete follow-up data on 1828 patients). 1110 (61%) of the patients were female, and mean age was 69 years (SD 10). The ES results are presented in Figs 1 and 2 and Tables 3 and 4. The

**Table 2. Demographics in patient subgroups.**

|  | Age: mean (SD) | Female (%) | BMI: mean (SD) | ASA ≥3 | Surgeons | Response coverage | N |
|---|---|---|---|---|---|---|---|
| Shoulder (WOOS) | 69 (10) | 62 | 29.5 (5.2) | 47 | 5 | 60% | 86 |
| Hip (OHS) | 68 (11) | 61 | 28.4 (4.5) | 36 | 20 | 72% | 747 |
| Knee (OKS) | 69 (9) | 61 | 29.6 (4.3) | 41 | 23 | 71% | 995 |
| **ALL** | **69 (10)** | **61** | **29.1 (4.4)** | **39** | **29** | **71%** | **1828** |

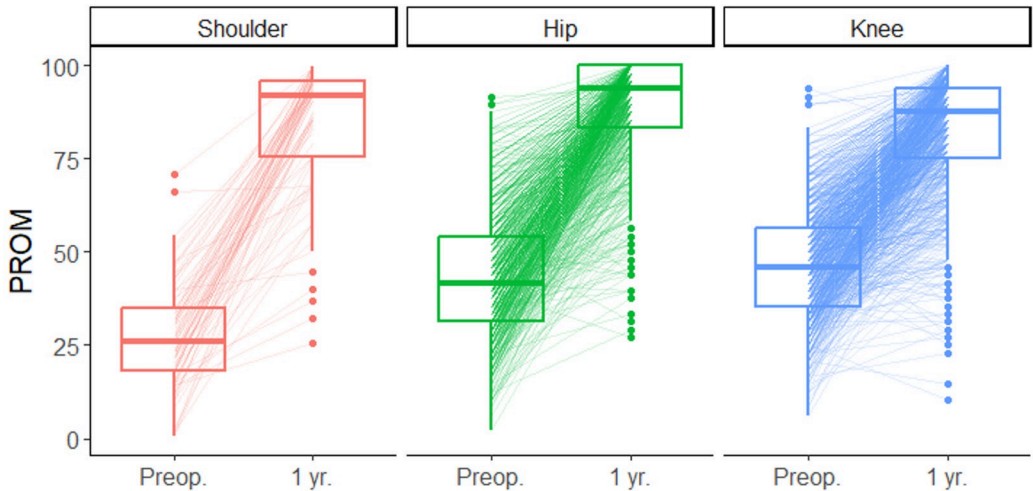

**Fig 1. PROM measurements for all patients included in the study before the operation (baseline) and at the 1-year follow-up mark.**

mean ES across all patient groups was 2.64 (SD 1.29) and the biggest mean ES was observed in shoulder patients (4.17, 95% C.I. [3.85, 4.49]), followed by hip patients (2.76, 95% C.I. [2.67, 2.84]), and the smallest in knee patients (2.42, 95% C.I. [2.34, 2.50]). The ES variation was highest among shoulder patients (SD 1.52), followed by knee patients (SD 1.27), and smallest for hip patients (SD 1.15). The ES was smaller for patients with higher preoperative PROM, with average ES being -0.051 (95% C.I. [-0.050, -0.052], p<0.0001) smaller per each additional preoperative PROM percentage point. Patients with higher ASA classes and BMI had on average lower ES, as did older patients. Patients' sex did not statistically significantly correlate with the ES. Hip and knee patients had lower ES than shoulder patients with the same preoperative PROM percentage scores.

The ES = 0.5 limits for responders based on this dataset were 6.8 (WOOS), 8.2 (OHS), and 7.4 (OKS). The percentage of responders was highest for shoulder patients (97.7%), followed by hip patients (96.8%), and lowest for knee patients (92.5%). There was a statistically significant difference between the number of responders for hip and knee surgeries (p<0.0001). However, when controlling for the preoperative PROM (%), the in between differences were not statistically significant (see S2 Table).

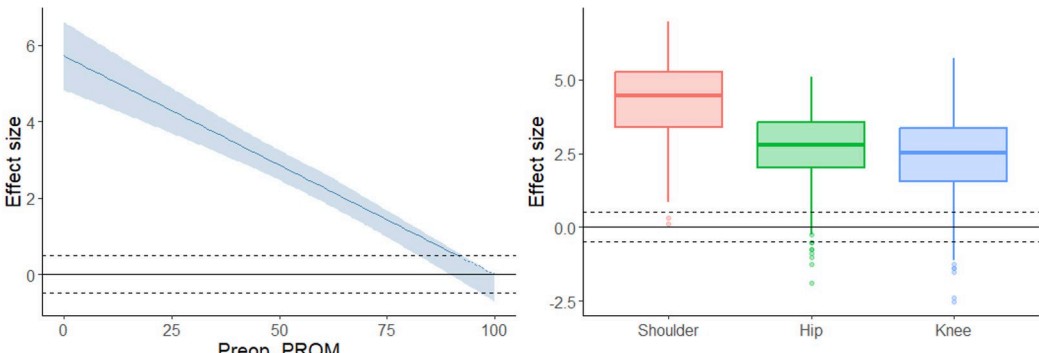

**Fig 2. Surgical outcomes measured as effect sizes for the patient groups included in the study.** Left plot for ASA = 2.

**Table 3. PROM scores (SD), mean changes and effect sizes (ES), their 95% confidence intervals and number of responders (%).** MCID from this data set: 6.8 (WOOS), 8.2 (OHS), and 7.4 (OKS).

| | Preop. | 1 yr. | Mean change | Effect size | Responders |
|---|---|---|---|---|---|
| Shoulder (WOOS) | 28 (14) | 83 (20) | 56.5 [52.1, 60.8] | 4.17 [3.85, 4.49] | 84 (97.7%) |
| Hip (OHS) | 43 (16) | 88 (14) | 45.2 [43.9, 46.5] | 2.76 [2.67, 2.84] | 723 (96.8%) |
| Knee (OKS) | 46 (15) | 82 (16) | 36.0 [34.8, 37.2] | 2.42 [2.34, 2.50] | 920 (92.5%) |

The differences in mean ES and unnormalized change in PROM were both highly statistically significant ($p < 0.0001$) between the anatomical groups. However, all ended up with high average one-year PROMs: 83 (WOOS), 88 (OHS), and 82 (OKS).

## Discussion

The primary finding of this study was a statistically significant difference in treatment ES among patients undergoing primary joint arthroplasty surgery for osteoarthritis in knee, hip, or shoulder. Accordingly, the number of responders and non-responders between patient groups was also significantly different between knee and hip patients. The lack of statistical significance when comparing to shoulder patients relates to the much smaller sample size, with 84 responders and only two non-responders. The mean ES were high and well above the estimated clinically significant levels in all joint arthroplasty groups.

Hip and knee arthroplasties are renowned for their high patient satisfaction [3], and although shoulder arthroplasties are less frequent, previous studies have reported similarly high satisfaction [11, 12]. A widely accepted threshold for a large ES in clinical medicine is 0.8 [13]. Within our study, we observed that the average ES among shoulder and hip patients was notably higher when compared to knee patients. Furthermore, the ES negatively correlated with ASA class, age, and especially with the preoperative PROM-scores. Interestingly, the one-year follow-up scores were essentially similar in all groups and the differences in ES were mostly due to variation in the baseline score. Accordingly, there seems to be a mean attainable PROM end-result limit (around 85%) for joint arthroplasty surgery. In order for a patient to subjectively benefit from the operation the baseline score has to be appropriately low, and our results allude to looser indications of knee arthroplasty surgery in terms of patient symptoms. It is noteworthy that all operations were performed due to end stage i.e. bone to bone osteoarthritis, and the radiographic findings do not always correlate with the clinical presentation [14–16].

**Table 4. Regression coefficients for a linear model with effect size (ES) as the response variable.**

| Variable | Estimate | SE | p-value |
|---|---|---|---|
| Intercept [a] | 5.943 | 0.131 | <0.0001 |
| Age$_{65}$ [b] | -0.010 | 0.004 | 0.008 |
| Preop. PROM (%) | -0.051 | 0.001 | <0.0001 |
| ASA | | | |
| II | -0.255 | 0.078 | 0.001 |
| III | -0.404 | 0.082 | <0.0001 |
| IV | -0.682 | 0.327 | 0.04 |
| Hip | -0.642 | 0.111 | <0.0001 |
| Knee | -0.803 | 0.110 | <0.0001 |

[a] Baselevel is "shoulder", age 65, ASA class I

[b] zero for ages 65 and below, Age–65 for older patients

Osteoarthritis represents a significant global disease burden [17]. Therefore, transparent reporting of health care outcomes may be regarded as a responsibility of all care providers [18]. PROMs can also be viewed as quality indicators for healthcare, given their ability to capture also patient-perceived adverse events and care-related complications. Nonetheless, one must bear in mind that changes in subjective health after a surgical intervention comprise of both specific and non-specific effects [19] and sometimes the latter might be of greater significance. Therefore, the ES in our study do not represent the differences in the effect of surgical intervention per se, but rather differences in the overall perceptions of our treated patients. Psychological factors are reported to play an important role in treatment outcome [19, 20].

We employed three disease-specific PROMs, selected by our specialists based on international joint arthroplasty collaboration [21, 22]. However, despite a shared denominator of primary osteoarthritis the PROMs are likely not fully comparable due to different psychometric properties [23]. The concepts of responder and MCID are also theoretical, in addition to being context and patient group dependent [24]. Despite the best efforts of, e.g., International Consortium of Health Outcome Measures (ICHOM) and consensus-based standards for the selection of health measurement instruments (COSMIN) there is a lack of uniform inclusive commitment [4, 25]. Furthermore, PROM validation should proposedly be an iterative cross-cultural process instead of a static designation that a PROM is valid [26]. Only 2 of our PROM questionnaires have been properly cross culturally validated in Finnish thus far, and the third (WOOS) is used as an institutionally translated version, yet without published data on psychometrics.

We acknowledge that in addition to potential criticism related to PROM selection, there are other weaknesses in our study. First, despite the analyses being controlled for ASA class and age [27], we could not control for all potential confounders, such as detailed radiographic findings, operative technique, and comorbidities. Second, PROMs are known to suffer from floor and ceiling effects [28]. Although, a marked ceiling effect was detected in the hip patients in our study (25%), this mostly had the effect of suppressing the high ES to some degree. Third, we lacked a fully comparable generic PROM or a uniform global score for all patients. Fourth, large divergence in same PROM MCID estimates between different patient populations and also statistical methods has been reported [29], and our responder analysis was based on distribution only.

As a strength, this was a systematic and comprehensive tertiary hospital-based registry overview with a large number of consecutive patients. We included fully completed data only, and given the number of patients, it is likely that the detected differences represent genuine variations between patient groups rather than individual fluctuations. One-year follow-up is also likely sufficient in capturing the biggest treatment effect [19].

After analyzing the results, we would like to emphasize the following points. Careful patient selection is essential for all patients undergoing artificial joint replacement, particularly those receiving knee arthroplasty. Our results suggest that knee arthroplasty outcomes are comparatively weaker than those in other major joint arthroplasties, potentially due to more lenient criteria for surgery in some cases. Preoperative PROM scores should be included in the decision making process. It is recommended to review both the patient's initial condition and the expected outcome prior to deciding on surgery, allowing any unrealistic expectations to be addressed. Additionally, patients should be informed that factors such as age, comorbidities and excess weight significantly impact the expected outcomes, regardless of the technical success of the procedure.

## Conclusion

The observed PROM ES for major joint arthroplasty surgeries is high. However, there are significant disparities among patients with primary knee, hip, and shoulder joint

arthroplasty surgery. Based on our observation the shoulder patients experience the largest treatment effect, and the knee arthroplasty patients the lowest. This variation underscores the importance of patient selection criteria, emphasizing also the need for thorough consideration of preoperative PROMs in the shared decision making regarding joint arthroplasty surgery. The expected results should be openly communicated to all stakeholders and further research targeted in improving PROM implementation, and treatment outcomes throughout orthopedics.

## Supporting information

**S1 File.**
(PDF)

**S2 File.**
(JPEG)

**S1 Table. Most common diagnoses and operations in patient groups.** Note that both numbers refer to the entire registry and thus the operation percentages do not refer only to the diagnosis on the same row of the table.
(DOCX)

**S2 Table. Regression coefficients for a generalized linear model with responder-status as a binary response variable.**
(DOCX)

**S3 Table. Regression coefficients for a linear model with change in PROM (%) as the response variable.**
(DOCX)

## Acknowledgments

The authors want to give thanks to Maria Suuripää (BCB Medical, Turku, Finland) for helping us with the data search.

## Author Contributions

**Conceptualization:** Ville Äärimaa, Fanny Kaivonen, Inari Laaksonen.

**Data curation:** Joel Kostensalo, Fanny Kaivonen.

**Formal analysis:** Joel Kostensalo.

**Investigation:** Fanny Kaivonen.

**Methodology:** Ville Äärimaa, Karita Kohtala, Fanny Kaivonen, Inari Laaksonen.

**Project administration:** Ville Äärimaa, Inari Laaksonen.

**Supervision:** Ville Äärimaa, Inari Laaksonen.

**Visualization:** Joel Kostensalo.

**Writing – original draft:** Ville Äärimaa, Inari Laaksonen.

**Writing – review & editing:** Ville Äärimaa, Karita Kohtala, Keijo Mäkelä, Mikko Karvonen, Anssi Arimaa, Anssi Ryösä, Inari Laaksonen.

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
