## [Decision Letter · Decision Letter 0]

25 Oct 2024

PONE-D-24-20361Comparative analysis of patient-reported outcomes in joint arthroplasty surgeriesPLOS ONE

Dear Dr. Kohtala,

Thank you for submitting your manuscript to PLOS ONE. After careful consideration, we feel that it has merit but does not fully meet PLOS ONE’s publication criteria as it currently stands. Therefore, we invite you to submit a revised version of the manuscript that addresses the points raised during the review process.

Please address all of the suggested edits and additions made by the reviewers. 

We look forward to receiving your revised manuscript.

Kind regards,

Joshua William Giles, Ph.D.

Academic Editor

PLOS ONE

“Ville Äärimaa has received grants from the Academy of Finland, the Social Insurance Institution of Finland, and the Turku University hospital. Ville Äärimaa has received payment for expert testimony given to National Patient Injury Board.

Inari Laaksonen have received grants from the State Research Funding of Southwestern Finland.

Turku University Hospital has supported Ville Äärimaa, and Inari Laaksonen to attend meetings and/or travel.”

Please respond by return email with your amended Competing Interests Statement and we will change the online submission form on your behalf.

3. In the online submission form you indicate that your data is not available for proprietary reasons and have provided a contact point for accessing this data. Please note that your current contact point is a co-author on this manuscript. According to our Data Policy, the contact point must not be an author on the manuscript and must be an institutional contact, ideally not an individual. Please revise your data statement to a non-author institutional point of contact, such as a data access or ethics committee, and send this to us via return email. Please also include contact information for the third party organization, and please include the full citation of where the data can be found.

Additional Editor Comments:

Thank you for preparing an interesting paper. Please implement the revisions suggested by the reviewers.

Reviewers' comments:

Reviewer's Responses to Questions

**Comments to the Author**

1. Is the manuscript technically sound, and do the data support the conclusions?

Reviewer #1: Yes

Reviewer #2: Yes

2. Has the statistical analysis been performed appropriately and rigorously? 

Reviewer #1: Yes

Reviewer #2: Yes

3. Have the authors made all data underlying the findings in their manuscript fully available?

Reviewer #1: Yes

Reviewer #2: Yes

4. Is the manuscript presented in an intelligible fashion and written in standard English?

Reviewer #1: Yes

Reviewer #2: Yes

5. Review Comments to the Author

Reviewer #1: The manuscript provides unique results regarding the enormous sample size of the joint arthroplasties. However, I suggest the authors add a new paragraph on clinical implications by interpreting the results. Also, these implications should be added to abstract conclusions. Discussion should be more extended to indicate these clinical implications.

Reviewer #2: Thank you for your valuable study. Addressing and interpretating the PROMs is not always as easy as it seems. Your study highlights important concerns when dealing with PROMS. Please make sure that the study followed the submission guideline of the PLoS One.

6. PLOS authors have the option to publish the peer review history of their article (what does this mean?). If published, this will include your full peer review and any attached files.

Reviewer #1: No

Reviewer #2: No

---

## [Author Response · Author response to Decision Letter 0]

13 Nov 2024

We have attached a new cover letter with this submission, detailing all the changes we have made.

Below are the most important sections of our response to reviewers file that we sent.

"Thank you for your comments and time spent on our article.

All changes made to the manuscript have been highlighted with blue font in the revised manuscript. 

We have copied below the comments and improvement suggestions of the reviewers, and provided point-by-point responses where necessary along with explanations of how we have modified our manuscript to address the raised issue. Our answers are colored with red font so that our answers are easily distinguishable from the comments of the editor and the referees.

5. Review Comments to the Author

Reviewer #1: The manuscript provides unique results regarding the enormous sample size of the joint arthroplasties. However, I suggest the authors add a new paragraph on clinical implications by interpreting the results. Also, these implications should be added to abstract conclusions. Discussion should be more extended to indicate these clinical implications.

Answer: Thank you for your comments and suggestions. In the original article, we covered the topic of clinical implications shortly in the conclusion paragraph, but now we have added a new, separate paragraph on the topic to the discussion section. The new paragraph is highlighted in blue in the revised manuscript and begins with: “After analyzing the results, we would like to emphasize…”

 We have also edited the abstract so that the clinical implications are mentioned by adding the end “We recommend that prior to shared decision-making, preoperative scores are thoroughly reviewed with the patient, along with other patient specific factors that may influence the end result of the treatment.”

Reviewer #2: Thank you for your valuable study. Addressing and interpretating the PROMs is not always as easy as it seems. Your study highlights important concerns when dealing with PROMS. Please make sure that the study followed the submission guideline of the PLoS One.

Answer: Thank you for your comments. We have gone through the journal’s submission guideline again carefully, and based on this, made the necessary changes to the article itself and to other materials to be submitted."

---

## [Editor Report · Decision Letter 1]

18 Nov 2024

Comparative analysis of patient-reported outcomes in joint arthroplasty surgeries

PONE-D-24-20361R1

Dear Dr. Kohtala,

We’re pleased to inform you that your manuscript has been judged scientifically suitable for publication and will be formally accepted for publication once it meets all outstanding technical requirements.

Kind regards,

Joshua William Giles, Ph.D.

Academic Editor

PLOS ONE
---

## [Editor Report · Acceptance letter]

10 Dec 2024

PONE-D-24-20361R1 

PLOS ONE

Dear Dr. Kohtala, 

I'm pleased to inform you that your manuscript has been deemed suitable for publication in PLOS ONE. Congratulations! Your manuscript is now being handed over to our production team.

Kind regards, 

on behalf of

Professor Joshua William Giles 

Academic Editor

PLOS ONE